# Assessing the Relation between Plasma PCB Concentrations and Elevated Autistic Behaviours using Bayesian Predictive Odds Ratios

**DOI:** 10.3390/ijerph16030457

**Published:** 2019-02-05

**Authors:** Brendan A. Bernardo, Bruce P. Lanphear, Scott A. Venners, Tye E. Arbuckle, Joseph M. Braun, Gina Muckle, William D. Fraser, Lawrence C. McCandless

**Affiliations:** 1Faculty of Health Sciences, Simon Fraser University, Burnaby, BC V5A 1S6, Canada; brendan_bernardo@sfu.ca (B.A.B.); bpl3@sfu.ca (B.P.L.); scott_venners@sfu.ca (S.A.V.); 2Population Studies Division, Environmental Health Science and Research Bureau, Healthy Environments and Consumer Safety Branch, Health Canada, Ottawa, ON K1A 0K9, Canada; tye.arbuckle@hc-sc.gc.ca; 3Department of Epidemiology, Brown University, Providence, RI 02912, USA; joseph_braun_1@brown.edu; 4École de psychologie, Université Laval, Québec, QC G1V 0A6, Canada; Gina.Muckle@psy.ulaval.ca; 5Department d’obstétrique et gynécologie, Université de Sherbrooke, Sherbrooke, QC J1H 5N4, Canada; William.Fraser@usherbrooke.ca

**Keywords:** autism, polychlorinated biphenyls, environmental chemicals, children, neuro-development

## Abstract

Autism spectrum disorder (ASD) is a neurodevelopmental condition characterized by impaired social communication and repetitive or stereotypic behaviours. In utero exposure to environmental chemicals, such as polychlorinated biphenyls (PCBs), may play a role in the etiology of ASD. We examined the relation between plasma PCB concentrations measured during pregnancy and autistic behaviours in a subset of children aged 3–4 years old in the Maternal-Infant Research on Environmental Chemicals (MIREC) Study, a pregnancy and birth cohort of 546 mother-infant pairs from Canada (enrolled: 2008–2011). We quantified the concentrations of 6 PCB congeners that were detected in >40% of plasma samples collected during the 1st trimester. At age 3–4 years, caregivers completed the Social Responsiveness Scale-2 (SRS), a valid and reliable measure of children’s reciprocal social and repetitive behaviours and restricted interests. We examined SRS scores as both a continuous and binary outcome, and we calculated Bayesian predictive odds ratios for more autistic behaviours based on a latent variable model for SRS scores >60. We found no evidence of an association between plasma PCB concentrations and autistic behaviour. However, we found small and imprecise increases in the mean SRS score and odds of more autistic behaviour for the highest category of plasma PCB concentrations compared with the lowest category; for instance, an average increase of 1.4 (95%PCI: −0.4, 3.2) in the mean SRS (exposure contrast highest versus lowest PCB category) for PCB138 translated to an odds ratio of 1.8 (95%PCI: 1.0, 2.9). Our findings illustrate the importance of measuring associations between PCBs and autistic behaviour on both continuous and binary scales.

## 1. Introduction

Autism spectrum disorder (ASD) is a neurodevelopmental condition affecting 1–2% of children that is characterized by impaired social communication and repetitive or stereotypic behaviours that manifest during early childhood [1]. It has been suggested that maternal exposure to some environmental chemicals during fetal development may play a role in the etiology of ASD [2,3,4,5,6,7,8]. The first and third trimesters of pregnancy have been identified as important developmental windows for chemical exposure [9]. One such class of chemicals is the polychlorinated biphenyls (PCBs), which have well-established neurotoxic properties [10]. PCBs have been banned in Canada since the late 1970s [11] and globally since 2004 [12]. Still, PCBs continue to persist in the environment [13]. Many studies have found that PCBs can affect mechanisms thought to be involved in the etiology of ASD, including immune response and functions, neuronal development, neuroexcitability, oxidative stress, and steroid hormones [8,14,15,16,17].

The impact of prenatal PCB exposure on neurodevelopment in children has been studied extensively [16,18,19,20,21,22]. However, only a few studies have examined PCBs in relation to ASD, and the effects of low-level PCB exposure are uncertain. Braun et al. [22] reported modest, but imprecise differences in autistic behaviours associated with gestational exposure to several endocrine disrupting chemicals, including some PCBs. A small case-control study by Cheslack-Postava et al. [20] reported weak associations between in utero PCB exposure and ASD, whereas no associations were seen in a larger follow-up sample from the same study [23]. More recently, in a large population-based case-control study, Lyall et al. [21] reported that several PCB congeners were associated with increased ASD risk in children. Overall, the effects of low-level PCB exposure on ASD remain unclear.

The purpose of this study was to examine the relation between plasma PCB concentrations measured during the first trimester of pregnancy and elevated autistic behaviour in 3- to 4-year-old children using the social responsiveness scale (SRS). We used Bayesian methods to analyze data from the Maternal Infant Research on Environmental Chemicals (MIREC) Study, a prospective cohort study of Canadian women and children. 

Bayesian methods have unique advantages in epidemiological studies with small effect sizes [24]. One advantage is that we can model uncertainty in population parameters, and functions of parameters (e.g., probability that θ > c), using the posterior distribution. This permits a more flexible and nuanced interpretation of the data compared to frequentist methods. In the present work, build on Gelman and Hill [25], and we propose a novel measure of association between exposure and disease called the Bayesian predictive odds ratio (BPOR). We use BPORs to examine SRS scores as both a continuous and binary outcome. The method uses a latent variable model for SRS >60, which assumes that all percentiles of the SRS distribution are equally affected by PCB exposure during pregnancy. A unique advantage of BPORs is they quantify how small shifts in the mean SRS between exposure levels translate into multiplicative changes in the odds of more autistic behaviour [21,26]. This enables a direct comparison of odds ratios based on the SRS, with odds ratios for clinical ASD taken from case-control studies such as in Lyall et al. [21].

## 2. Materials and Methods 

### 2.1. Maternal-Infant Research on Environmental Chemicals (MIREC) Study

We used data from the MIREC study, a prospective pregnancy and birth cohort study of 2001 women from ten Canadian cities between 2008 and 2011. The goal of the MIREC Study was to obtain national biomonitoring data on pregnant women and their infants to examine the effects of prenatal exposure to environmental chemicals on pregnancy and child health outcomes [27]. Study criteria and further details about participant eligibility and exclusions are discussed in the cohort profile by Arbuckle et al. [27]. For this study, we employed the subsample of participants in the MIREC follow-up neurodevelopment study when the children were 3 to 4 years old (average: 3.4 years). We included mothers who had socio-demographic and child neurodevelopment information, as well as plasma PCB concentrations and total lipid concentrations measured during the first trimester of the pregnancy. A total of 546 met all the above criteria for inclusion in our analysis. This research was approved by ethics review boards from Health Canada, Sainte-Justine Research Center, and Simon Fraser University. All women provided informed consent for their and their child’s participation in the study.

### 2.2. Biomarkers of PCB Exposure 

We measured concentrations of 24 congeners in plasma samples collected during the first trimester of the pregnancy, at an average of 12.0 weeks gestation (range: 6.0–14.0 weeks). Biomarker analysis occurred at the Toxicology Laboratory of the Institut national de santé publique du Québec, and all samples were stored at −20 °C [28]. We quantified PCB congener concentrations (International Union for Pure and Applied Chemistry nos. 28, 52, 66, 74, 99, 101, 105, 118, 128, 138, 146, 153, 156, 163, 167, 170, 178, 180, 183, 187, 194, 201, 203, 206) using gas chromatography/mass spectrometry [28]. We retained the six PCB congeners that were detected in at least 40% of samples. Measurements below the LOD were replaced using the single imputation “fill-in” approach where the log-PCB concentrations <LOD were randomly sampled from a truncated lognormal distribution with mean and standard deviation estimated from the observed data [29]. The “fill-in” approach for missing biomarkers yields unbiased regression coefficient estimates if the imputation distribution is correct, although standard errors may be biased. We also calculated the sum of the six PCBs, weighted by molar mass, to estimate the relation between combined exposure to multiple PCBs and the SRS score. We did not consider summations weighted using toxic equivalency factor (TEF) calculations, because the only dioxin-like congener that was detected in >40% of samples was PCB118. Axelrad et al. [30] examined different PCB body burden metrics and recommended using the sum of the most frequently detected congeners. To account for individual-level variability in plasma lipid levels, we standardized all PCB concentrations by total plasma lipid concentrations and expressed in units of ng/g lipids [31,32].

### 2.3. Social Responsiveness Scale Score

The Social Responsiveness Scale-2 (SRS-2) was the dependent variable in our analysis, a valid and reliable caregiver-reported questionnaire that provides a quantitative measure of autistic behaviour and, at higher scores, differentiates autism from other disorders [33]. The SRS score has been cross-validated in a large European sample of clinical ASD cases [34], and it has been compared with the Diagnostic and Statistical Manual of Mental Disorders (DSM) [35]. The SRS consists of a series of 65 questions on a Likert Scale that measure a child’s behavioural characteristics during the previous 6 months. The sum of the questions gives a total T-score, where higher scores describe greater deficiencies in reciprocal social behaviour (i.e. interpersonal, repetitive, or stereotypic behaviours) that are more likely to indicate clinically diagnosed autism spectrum disorder [34,36,37]. SRS score cut-offs have been defined to denote the range of autistic behaviours. Scores from 60 to 65 are categorized as ‘Mild’, 66 to 75 as ‘Moderate’, and above 75 as ‘Severe’. The SRS has two DSM-V subscales for ASD and five treatment subscales (social awareness, social cognition, social communication, social motivation, and restricted interests and repetitive behaviour), which measure receptive, cognitive, expressive and motivational aspects of social behaviour as well as autistic preoccupations.

### 2.4. Covariates

We included variables that may potentially confound the relationship between plasma PCB concentrations and autistic behaviours. We created a Directed Acyclic Graph (Figure 1) to identify factors that were either predictors of autistic behaviour, or alternatively, common causes of PCB exposure and autistic behaviour [38]. We excluded breastfeeding and pregnancy outcomes from our models for SRS because they may be affected by PCB levels [39,40], and therefore, mediating variables on the causal pathway between exposure and outcome. We also excluded biomarkers of other prenatal contaminant exposures, including lead, because a previous study in the MIREC Study cohort found little evidence of associations with cognitive function [41]. Our final set of covariates included: child sex, mother’s age in years, maternal race (white, other), maternal education (four levels), annual income (four levels), marital status (married, other), ever smoked/consumed alcohol during pregnancy, and pre-pregnancy BMI (four levels). We used the same set of covariates in all analyses.

### 2.5. Analytic Approach

We used Bayesian linear regression to estimate the confounder-adjusted associations between plasma PCB concentrations and mean SRS score. We designed our analysis to make direct comparisons with the analysis results of Lyall et al. [21]. Each PCB concentration was included in a single-pollutant model for SRS using indicator variables to create four categories of PCB exposure, along with measured covariates. The categories were the PCB quartiles used in Table 1 of Lyall et al. [21]. Because categorization of a continuous exposure subject to measurement error can induce non-differential misclassification [42], we also examined log 2 transformed PCB as a continuous exposure in linear regression analysis, and additionally, we created scatter-plots of each PCB versus SRS.

To illustrate the Bayesian approach, let *Y* be the SRS score, X be a single PCB exposure variable (e.g., PCB138), and C = {C1, C2, …, CK} be the vector of K confounders. Define Q1, Q2, Q3, Q4 to be zero-one indicator variables for inclusion in the 1st, 2nd, 3rd or 4th PCB categories, which were the four quartiles from Table 2 of Lyall et al. [21]. For example, for PCB138, the quartiles were [0,2.9), [2.9,4.2), [4.2,6.2), and [6.2,46.8] measured in ng/g lipids. We modelled the outcome Y using multiple linear regression:
(1)$$Y=\beta_{1}+\beta_{2}Q_{2}+\beta_{3}Q_{3}+\beta_{4}Q_{4}+\beta_{C_{1}}C_{1}+\beta_{C_{2}}C_{2}+\ldots +\beta_{C_{K}}C_{K}+\varepsilon$$
where ε ~ Logistic (0, λ), which is linear regression with indicator variables for each category, using Q1 as the reference category. We use logistic errors rather than normally distributed errors, which are usually used in linear regression. The reason is because logistic errors enable a logistic regression interpretation of Equation 1 based on a latent variable formulation. As described below and in [25] logistic errors imply that the log odds of Y>c is a linear equation, but this is not true for normally distributed errors.

We did not adjust for co-pollutant confounding from multiple PCBs in Equation 1 because the biomarkers are highly correlated, which induces variance inflation in parameter estimates. For the parameters *β*_2_, *β*_3_, *β*_4_, which describe the association between PCB exposure category and mean SRS, adjusted for confounders, we assigned an autoregressive prior to smooth the dose-response curve, where:(2)$$\beta_{i+1}∼N\left(\beta_{i},\tau^{2}\right)\;\mathrm{for}\;i=1,2,3$$
and we assigned:(3)$$\tau^{2}∼N\left(0,10000\right),\tau^{2}>0.$$

The parameter $\tau^{2}$ induces a shrinkage factor that pulls the segments of the dose-response curve together. This approach gives a more realistic dose-response curve, including smaller 95% intervals, better prediction of Y, and is less susceptible to random errors in the data [43]. Additionally, we assigned uninformative priors to the parameters $\beta_{1},\beta_{C_{1}},\beta_{C_{2}},\ldots ,\beta_{C_{K}},\delta^{2}$:(4)$$\beta_{1},\beta_{C_{1}},\beta_{C_{2}},\ldots ,\beta_{C_{K}}∼N\left(0,10000\right);\lambda ∼N\left(0,10000\right),\lambda >0.$$

We used Markov Chain Monte Carlo (MCMC) with the software Stan [44] to generate a sample from the posterior distribution [45].

To make direct comparisons with the odds ratios for ASD observed in Lyall et al. [21], we examined SRS scores as both a continuous and a binary outcome. We calculated BPORs for more autistic behaviour based on a threshold SRS score >60 using the MCMC output. To compute the odds of more autistic behaviour (SRS>60) for participants in the uppermost PCB category divided by the odds of more autistic behaviour in the lowest PCB category, we computed:(5)$$BPOR=\frac{odds\left(Q4=1\right)}{odds\left(Q1=1\right)}=\frac{1-plogis\left(60,\beta_{1}+\beta_{4},\lambda \right)}{1-plogis\left(60,\beta_{1},\lambda \right)}$$
where the values of $\beta_{1},\beta_{4},\lambda$ are MCMC sample iterations from the posterior distribution in Equation (1). The quantity 1 – plogis (60, *β*, λ) is the probability that SRS >60, and plogis(.) is the cumulative distribution function of a logistic random variable.

Because of the properties of the logistic distribution, the value of the BPOR (e.g., for Q4 versus Q1) does not depend on the particular threshold c that is used to define more autistic behaviour. The reason is because if the mean of a logistic random variable is μ=α+βX, then the odds that Y>c is given by:(6)$$Odds(Y>c|\mu ,\lambda )\;=\;exp\left\{-\left(c-\mu \right)/\lambda^{}\right\}=exp\left\{-\left(c-\left(\alpha +\beta X\right)\right)/\lambda \right\},$$
and therefore, the odds ratio of Y>c based on a unit increase in X, is given by
(7)Odds(Y>c|X=x+1,α,β,λ)/Odds(Y>c|X=x,α,β,λ) =exp(β/λ),
which does not depend on c. Therefore, with no loss of generality, we use a threshold of SRS > 60 in to compute BPORs in the MIREC Study.

We obtained 95% posterior credible intervals (PCIs) for model parameters, which are the Bayesian equivalent of frequentist confidence intervals (CIs). The abbreviation CI is specific to frequentist confidence intervals. The parameters are interpreted as random variables and the boundaries indicate a 95% probability range for the unknown quantity. The BPOR approach gives much narrower 95% interval estimates for odds ratios compared to logistic regression directly on the dichotomized SRS score because it uses the underlying linear model for SRS to predict the probability that SRS > 60. Additionally, we used the MCMC samples to calculate the posterior probability that the BPORs were greater than 1.0.

We conducted additional multiple linear regression analyses to estimate differences in subscales of the SRS score (Social Awareness, Social Cognition, Social Communication, Social Motivation, and Restricted Interests and Repetitive Behaviour). We conducted a sex stratified analysis for males and females because ASD is more prevalent in boys and the effects of PCB exposure during pregnancy may differ by sex [3,46,47,48].

## 3. Results

### 3.1. Descriptive Statistics

The women were generally ≥30 years of age (77.5%) with a post-secondary degree (94.7%), and had an annual household income ≥$80,000 (59.0%) (Table 3). Eight (six boys and two girls) out of 546 children in the MIREC sample (1.5%) had SRS scores > 60. Characteristics that were predictive of lower SRS scores included: higher maternal age, higher education, an annual income ≥$100,000, and having a female child.

Six (118, 138, 153, 170, 180, 187) of 24 PCB congeners were detected in >40% of participants; PCB153, which was the most prevalent, was detected in 100% of participants (Table 3). Geometric mean concentrations (ng/g lipid) among the six PCBs were as high as 7.9 (range: 1.7–80.9) for PCB153 and as low as 1.2 (range: 0.5–26.9) for PCB187. The geometric means of PCBs for pregnant women in MIREC were generally lower than the Canadian women of childbearing age (20–39 years) from Canadian Health Measures Survey (CHMS) [49,50]. They were also lower than levels reported in an earlier survey of chemical exposures in pregnant American women [51].

The Cronbach’s alpha for all SRS scores (i.e., total SRS, two DSM-V subscales and five treatment subscales) were all greater than 0.90, which indicates a high internal consistency. To calibrate our inferences about the strength of the association between plasma PCB exposure and SRS, we used frequentist regression analyses to show the relation between each participant characteristic and SRS score (Appendix A). Children who were male and had mothers who were not married had higher mean SRS scores, with mean differences in SRS score of 2.3 (95% CI: 1.3, 3.3) versus female and 1.5 (95% CI: 0.3, 2.7) versus married mothers, respectively.

### 3.2. Linear Regression Analyses

Higher plasma PCB concentrations were associated with subtle non-monotonic differences in continuous SRS scores, meaning that we saw a change in direction of the response between the categories of ng/g lipid PCB concentrations (Table 1). All six PCBs, and the sum of PCBs, were associated with weak and imprecise increases (large posterior standard deviation) in the mean SRS when comparing the fourth category with the first category. For instance, mean SRS scores were 1.4 points (95%PCI: −0.4, 3.2) higher among children in the 4th PCB138 category compared to the 1st category. In other settings, comparisons of the 2nd or 3rd PCB category to the lowest category were associated with reductions in the mean SRS score (e.g., PCB180 in Table 1). Appendix A examines PCBs as a continuous exposure, and the results illustrate a pattern of associations, influenced by confounding bias from participant characteristics, wherein higher PCBs levels are associated with imprecise increases in the mean SRS.

### 3.3. BPOR Analyses

In the BPOR analyses of Table 4, for all six PCBs, we observed small increases in the odds of SRS > 60 in the fourth category compared to the first category, although the resulting 95% interval estimates were very imprecise (large posterior standard deviation). For instance, PCB138 exhibited a modest odds ratio in the fourth category compared to the first category, with an odds ratio of 1.76 (95%PCI: 0.99, 2.92). We also reported the posterior probabilities that the OR is greater than 1.0 (Table 4). This tells the reader that if the model is correct, then for PCB138, there is an estimated 98% probability that the odds ratio (fourth category versus first category) is greater than 1, emphasizing the value of the BPOR approach and narrower 95% interval estimates. To enable comparisons with previous research, Table 4 also includes odds ratios for ASD from Lyall et al. [21], and the results show some similarity with the BPORs from the MIREC Study.

To further demonstrate the advantage of Bayesian methods and the BPOR approach, we also calculated odds ratios for more autistic behaviour (SRS > 60) using frequentist logistic regression directly on the dichotomized SRS scores (Table 4). A total of eight (1.5%) out of 546 children (six boys and two girls) had SRS > 60. Because only 1.5% had scores for SRS > 60, we see that the ORs from a traditional logistic regression model are extremely imprecise and, in many cases, the maximum likelihood estimator did not converge. In contrast, the 95%PCIs from BPOR are much narrower because they leverage the underlying linear regression model for SRS to provide more accurate estimation of the probability of more autistic behaviour (SRS > 60).

### 3.4. Supplemental Analyses of SRS Subscales and Stratification by Sex

We conducted additional analyses to estimate differences in subscales of the SRS score (Appendix A) and a sex stratified analysis for males (n = 261) and females (*n* = 285) (Appendix A). However the width of the 95% CIs for model parameters were wider, and this makes the interpretation of results more challenging. Changes in PCB concentrations were associated with small increases or decreases in the SRS subscales. In some cases, we saw the same patterns with the Bayesian models for the total SRS score (Appendix A). For instance, the Social Communication subscale had a mean increase of 1.8 [95%CI: 0.0, 3.6] for PCB138, compared to the first category. In the sex stratified analysis for PCB138, we saw an overall increase in the mean SRS score, with increases of 1.8 [95%PCI: −1.2, 4.8] for boys and 0.6 [95%PCI: −1.6, 2.8] for girls (Appendix A).

## 4. Discussion

We examined the relationship between plasma PCB concentrations and elevated autistic behaviour using a novel Bayesian analytic approach. We found no evidence of an association between plasma PCB concentrations and autistic behaviour. SRS scores differed modestly as plasma PCB concentrations increased; the associations were generally non-monotonic. However, all six PCBs, and the sum of PCBs, were associated with weak and imprecise increases in the mean SRS when comparing the highest PCB category with the lowest category. Comparisons of the 2nd or 3rd PCB category to the lowest category were in some cases associated with reductions in the mean SRS score (e.g., PCB180). Additionally, the BPORs provided an alternative framework to examine the SRS as a dichotomous outcome. We observed higher odds of elevated autistic behaviour in the highest PCB category compared with the lowest category, and additionally, we observed similarities between the BPORs in MIREC versus the ORs for ASD from Lyall et al. [21].

Interestingly, the 95% PCIs from BPORs were similar in size than the corresponding 95% CIs from Lyall et al. [21], even though only eight out of 546 children in the MIREC sample (1.5%) had scores for SRS > 60. The reason is because BPORs leverage the underlying linear regression model for SRS from Equation (1) to enable more accurate Bayesian predictions of the odds ratio compared to logistic regression directly on the dichotomized SRS score data. For example, in Table 4 for PCB138 (Q4 versus Q1) the OR is 1.76 [95%PCI: 0.99, 2.92] for the BPOR, which is remarkably similar to the OR from Lyall et al. [21] given by 1.79 [95%PCI: 1.10, 2.92].

An important finding is that small changes in the mean SRS score translate to observably larger changes in the odds of autistic behaviour based on an SRS threshold of 60. For example, for PCB138 an average increase of 1.4 [95%PCI: −0.4, 3.2] in the mean SRS for (Q4 versus Q1) translates to an odds ratio of 1.76 [95%PCI: 0.99, 2.92]. Furthermore, an important property of the logistic error model given in Equation (1), as detailed in Methods, is that the BPOR is invariant to the choice of threshold (e.g., 60 or 75) used to define more autistic behaviour [52]. These findings have important implications in the study of autistic behaviour because seemingly small shifts in the distribution of SRS can translate into larger effect sizes on the multiplicative risk scale. This phenomenon has been described in other studies looking at the impact of toxicants on children, as depicted in a YouTube video by Lanphear about how “Little things matter” [26,53,54,55].

This study builds on the existing literature examining in utero PCB exposure and autistic behaviours [16,18,19,20,21,22]. Cheslack-Postava et al. [20] found some evidence that higher total PCB levels were associated with high frequency of ASD, whereas Braun et al. [22] found evidence that several PCBs (e.g., PCB138) were associated with more autistic behaviours. The larger case-control study of Lyall et al. [21] presented clearer evidence of monotonic dose-response relationships between PCBs (e.g., PCB 138 and 153) and risk of ASD in offspring. In addition to differences in statistical methodology, differences in mixtures to which study populations are exposed (for example, other chemicals acting as confounders, or dietary factors that modify associations that vary across populations) could also account for discrepancies across existing work. More generally, there is evidence from human and animal studies that some PCB congeners are associated with neurotoxic endpoints even at low doses [56,57].

The study has several limitations. Although the linear model in Equation (1) accounts for non-linear dose-response using PCB quartiles, it is overly simplistic and does not adjust for co-pollutant confounding or interaction between PCB congeners. PCB concentrations by lipid volume may not reflect the dose in the target tissues, and categorization of the biomarker into categories can induce differential misclassification even if the underlying measurement error is non-differential [42]. We also examined PCBs as a continuous variable in the Appendix A (see Appendix A and Appendix A). Our analysis ignored the effect of combined exposure to multiple PCBs on autistic behaviour. Calculating the sum of PCBs is not satisfactory to characterize the combined effect of multiple PCBs. Individual congeners or the sum of PCBs with similar chemical properties may play a larger role in associations with autistic behaviour or other neurodevelopmental disorders [21]. Instead, we could have considered other Bayesian methods (e.g., [58,59]) which incorporate several PCBs into the same model, while imputing PCBs that fall below the LOD [60,61].

Another limitation of our analysis is that SRS scores are not a perfect quantitative measure of autistic traits. Our BPOR analysis defines elevated autistic behaviour using an established threshold of >60, which indicates mild, moderate or severe autistic behaviour [34], rather than clinical information. The SRS is not a diagnostic test for ASD; it may also capture other aspects of social behaviours and latent traits that tend to co-occur with other behavioural disorders such as attention deficit hyperactivity disorder or language disorders [34,62,63]. Furthermore, SRS may have low specificity for ASD, because of traits related to social motivation and ADHD [62,63,64]. Further study is needed to compare ORs based on the SRS with ORs based on ASD diagnosis.

## 5. Conclusions

In conclusion, this is one of only a few studies to examine in utero PCB exposures and autistic behaviour in a prospective cohort of pregnant women. We found no association between plasma PCB concentrations and autistic behaviour. However, we found small and imprecise increases in the mean SRS score and odds of more autistic behaviour for the highest category of plasma PCB concentrations compared with the lowest category. Our findings demonstrate the value of measuring associations between PCBs and autistic behaviour on both continuous and binary scales using Bayesian statistics. Further research is needed to examine the effects of chemical mixtures and combined exposure to multiple PCBs to improve our understanding of the effects of multiple correlated exposures.

## Figures and Tables

**Figure 1 ijerph-16-00457-f001:**
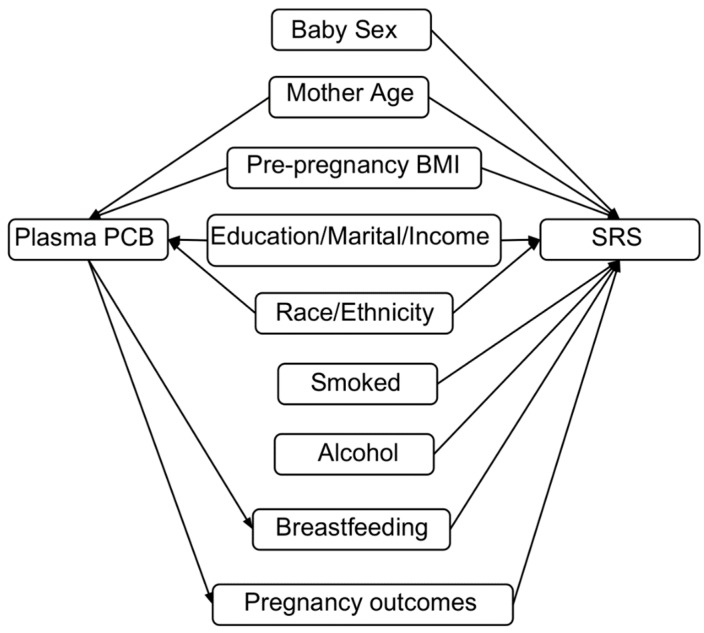
Directed Acyclic Graph (DAG) for the relation between plasma PCB exposure during pregnancy, the SRS score, and participant variables.

**Table 1 ijerph-16-00457-t001:** Mother PCB levels (quartiles) in relation to mean child SRS score in MIREC study participants, Canada, 2008–2011 using Bayesian linear regression (*n* = 546).

PCB Category ^1^	Value (ng/g Lipid)	*n*	SRS Unadjusted Mean Scores (95% CI)	SRS Adjusted ^2^ Mean Scores (95% CI)
**PCB118**				
Q1	<1.4	108	0.0 (referent)	0.0
Q2	1.4 -< 2.3	143	−0.03 (−1.49, 1.50)	0.09 (−1.46, 1.63)
Q3	2.3 -< 3.6	170	−0.49 (−1.90, 0.98)	−0.02 (−1.55, 1.53)
Q4	≥3.6	125	−0.36 (−1.89, 1.20)	0.26 (−1.34, 1.88)
**PCB138**				
Q1	< 3.2	175	0.0	0.0
Q2	3.2-< 5.5	184	0.10 (−1.13, 1.32)	0.70 (−0.63, 2.04)
Q3	5.5-< 8.9	118	−0.21 (−1.59, 1.18)	0.44 (−1.11, 2.01)
Q4	≥ 8.9	69	0.52 (−1.15, 2.19)	1.35 (−0.42, 3.16)
**PCB153**				
Q1	< 4.2	87	0.0	0.0
Q2	4.2-< 7.4	178	0.41 (−1.14, 1.95)	0.58 (−1.02, 2.19)
Q3	7.4-< 11.7	144	−1.08 (−2.70, 0.50)	−0.50 (−2.25, 1.26)
Q4	≥ 11.7	137	0.16 (−1.46, 1.76)	1.10 (−0.71, 2.89)
**PCB170**				
Q1	< 1.5	227	0.0	0.0
Q2	1.5-< 2.6	141	−0.79 (−2.04, 0.48)	−0.33 (−1.66, 1.02)
Q3	2.6-< 4.3	110	−1.12 (−2.49, 0.24)	−0.14 (−1.64, 1.33)
Q4	≥ 4.3	68	0.02 (−1.58, 1.64)	0.83 (−0.97, 2.62)
**PCB180**				
Q1	< 3.4	154	0.0	0.0
Q2	3.4-< 6.1	182	−1.99 (−3.25, −0.72)	−1.57 (−2.93, −0.16)
Q3	6.1-< 10.4	120	−2.00 (−3.41, −0.58)	−1.13 (−2.75, 0.50)
Q4	>= 10.4	90	−0.48 (−2.02, 1.05)	0.19 (−1.60, 1.97)
**PCB187**				
Q1	< 0.92	197	0.0	0.0
Q2	0.92-< 1.8	124	−0.30 (−1.64, 1.04)	−0.49 (−1.83, 0.88)
Q3	1.8-< 3.3	135	−0.86 (−2.15, 0.44)	−0.46 (−1.84, 0.94)
Q4	>= 3.3	90	−0.20 (−1.71, 1.27)	0.51 (−1.15, 2.15)
**Sum of above PCBs**				
Q1	< 33.4	358	0.0	0.0
Q2	33.4-< 55.3	110	−0.29 (−1.58, 0.98)	0.60 (−0.75, 1.96)
Q3	55.3-< 86.3	51	0.16 (−1.59, 1.93)	0.67 (−1.21, 2.53)
Q4	≥ 86.3	27	0.73 (−1.66, 3.12)	1.45 (−0.98, 3.90)

^1^ The Q1, Q2, Q3, Q4 are the 1st, 2nd, 3rd or 4th PCB quartiles from Table 1 of Lyall et al. [21]. ^2^ Adjusted for child’s sex, mother’s age, race, marital status, education level, annual income, whether the mother has ever smoked during pregnancy, has ever consumed alcohol during pregnancy, and pre-pregnancy BMI.

**Table 2 ijerph-16-00457-t002:** Distributions of Blood Plasma PCBs (ng/g lipid) during the first trimester for MIREC study participants, Canada, 2008–2011 (n = 546).

						MIREC					
Congener	%>LOD CHMS ^1^	%>LOD MIREC	GM ^2^ CHMS ^1^	GM MIREC	Mean MIREC	SD	25th	50th	75th	95th	Max
PCB118	83.2	77.5	3.09	2.1	2.9	2.6	1.7	2.4	3.4	6.9	30.2
PCB138	96.1	95.2	5.46	4.3	5.6	5.2	2.9	4.2	6.2	14.4	46.8
PCB153	91.6	100	8.22	7.9	10.1	9.7	4.9	7.5	11.7	25.0	80.9
PCB170	50.2	56.8	NA	1.4	2.6	3.5	0.7	1.9	3.1	7.2	40.3
PCB180	95.4	97.1	5.79	5.3	7.5	9.3	3.2	5.1	8.2	19.6	114.9
PCB187	41.1	46.0	NA	1.2	2.0	2.4	0.6	1.4	2.5	5.5	26.9
Sum of PCBs ^3^	NA	NA	NA	26.7	34.9	34.9	16.5	25.3	40.9	81.9	345.3

^1^ Plasma concentrations (ng/g lipid) for Canadian women of childbearing age (20–39 years), Canadian Health Measures Survey (CHMS) Cycle 1, 2007–2009 [49,50]. ^2^ GM = Geometric Mean (not calculated in CHMS when %>LOD was less than 60%). ^3^ Sum of PCBs 118, 138, 153, 170, 180, and 187 weighted by molar mass.

**Table 3 ijerph-16-00457-t003:** Sociodemographic characteristics of MIREC study participants, Canada, 2008–2011 (*n* = 546).

	n (%)	SRS (Median (IQR))	PCB118	PCB138	PCB153	PCB170	PCB180	PCB187	Sum of PCBs ^1^
			(ng/g Lipid) (Median (IQR))						
**Total**	546 (100)	44 (41–49)	2.4 (1.7–3.4)	4.2 (2.9–6.2)	7.5 (4.9–11.7)	1.9 (0.7–3.1)	5.1 (3.2–8.2)	1.4 (0.6–2.5)	25.3 (16.5–40.9)
**Child Sex**									
Male	261 (47.8)	45 (42–50)	2.5 (1.7–3.4)	4.3 (3–6.3)	7.6 (5–11.4)	1.8 (0.7–3)	5.1 (3.2–7.9)	1.4 (0.6–2.5)	26.5 (16.9–38.9)
Female	285 (52.2)	43 (40–47)	2.4 (1.6–3.4)	4.2 (2.8–6.2)	7.3 (4.8–11.8)	1.9 (0.7–3.1)	5.2 (3.2–8.4)	1.4 (0.6–2.5)	24.5 (15.5–41.6)
**Mother’s Age**									
19–29	122 (22.3)	45 (42–52)	1.9 (1.2–2.6)	3 (2.2–4)	5 (3.6–7.4)	1 (0.4–1.9)	3.1 (2.2–5.1)	1 (0.5–1.6)	16.7 (12.2–24.4)
30–34	205 (37.5)	44 (41–48)	2.3 (1.7–3.3)	4.2 (2.9–5.8)	7.2 (4.9–10.4)	1.5 (0.6–2.7)	4.8 (3.2–7.2)	1.2 (0.5–2.1)	24.1 (16.6–35.1)
35+	219 (40.0)	44 (40–47)	2.8 (2.1–4.2)	5.5 (3.6–7.8)	9.6 (6.7–14.3)	2.4 (1.6–3.8)	6.7 (4.8–10.3)	2.1 (1–3.3)	33 (23.5–49.5)
**Race**									
White	491 (89.9)	44 (40–49)	2.9 (2.4–4.8)	5.8 (3.7–9.1)	11.6 (7.1–18)	2.6 (1.9–5.4)	7.1 (5.1–12.7)	2.9 (1.2–3.8)	40.2 (25.9–57.6)
Other	55 (10.1)	44 (40–49)	2.9 (2.4–3.4)	6 (4.1–9.2)	13 (8.2–18)	3.5 (2–5.6)	9.8 (5.2–14)	3.1 (1.2–6.3)	46.5 (28.9–76.4)
**Marital Status**									
Married	241 (89.9)	44 (40–49)	2.9 (2.4–4.8)	5.8 (3.7–9.1)	11.6 (7.1–18)	2.6 (1.9–5.4)	7.1 (5.1–12.7)	2.9 (1.2–3.8)	40.2 (25.9–57.6)
Other	154 (28.2)	44 (40–49)	2.9 (2.4–4)	5.3 (3.7–9.2)	9.1 (6.9–18.2)	2.6 (1.9–5.5)	7 (5.1–13.2)	2 (1–5.7)	30.7 (25.9–70)
**Education Level**									
High School Diploma or less	29 (5.3)	44.5 (42–52.2)	1.4 (0.5–2.2)	2.6 (1.9–3.5)	4.6 (3.5–6.1)	0.7 (0.3–1.7)	3.1 (1.9–4.5)	0.6 (0.2–1)	14.5 (12.2–19.5)
College or Trade School Diploma	154 (28.2)	45 (42–50)	2.2 (1.5–3.1)	3.6 (2.5–5.8)	6 (4.2–9.9)	1.3 (0.4–2.5)	4 (2.6–6.7)	1.2 (0.5–2.3)	19.5 (14.1–33.8)
Undergraduate University Degree	213 (39.0)	45 (41–49)	2.4 (1.8–3.4)	4.4 (3–6.2)	7.5 (5–11.2)	1.8 (0.8–3)	5.1 (3.4–7.6)	1.4 (0.6–2.3)	24.9 (16.9–39)
Graduate University Degree	150 (27.5)	43 (40–47)	2.9 (2.1–3.9)	5.1 (3.6–7.3)	9.6 (6.9–13.2)	2.5 (1.6–3.6)	6.8 (4.8–9.9)	2 (1–3.3)	33.4 (23.7–47.9)
**Annual Household Income**									
≤$40,000	73 (13.4)	45 (42.8–52.2)	2.2 (1.5–3.1)	3.4 (2.4–5.8)	6.3 (3.7–11.2)	1.5 (0.6–2.8)	4.5 (2.3–7.4)	1 (0.4–2.3)	20.1 (12.8–39.1)
$40,001–$80,000	151 (27.7)	45 (41.5–50.5)	2.3 (1.7–3.5)	3.8 (2.7–6)	6.9 (4.5–10.5)	1.5 (0.5–2.7)	4.7 (2.9–7.2)	1.4 (0.8–2.4)	23.5 (14.9–36.2)
$80,001–$100,000	105 (19.2)	44.5 (40.8–49)	2.1 (1.2–3.1)	3.7 (2.7–5.9)	6.2 (4.6–10.7)	1.5 (0.7–2.7)	4.2 (3.2–7.6)	1.1 (0.4–1.9)	19.9 (14.8–36.6)
>$100,000	217 (39.7)	44 (40–47)	2.6 (1.9–3.9)	4.9 (3.5–6.8)	8.6 (5.9–12.5)	2.1 (1.2–3.5)	6 (4.1–9.6)	1.8 (0.7–2.9)	29.4 (19.5–44.6)
**Has Ever Smoked During Pregnancy**									
Yes	189 (34.6)	45 (40–49)	2.4 (1.6–3.4)	4.6 (2.9–6.4)	7.8 (5.2–12.5)	2 (0.8–3.4)	5.3 (3.3–8.9)	1.6 (0.6–2.7)	27 (17–42.4)
No	357 (65.4)	44 (41–48)	2.4 (1.7–3.4)	4.2 (2.9–6.1)	7.4 (4.9–11.3)	1.8 (0.7–3)	5.1 (3.2–8)	1.3 (0.6–2.4)	24.9 (16–38.6)
**Has Ever Consumed Alcohol During Pregnancy**									
Yes	91 (16.7)	44 (40–48)	2.7 (2.1–3.8)	4.5 (3.4–6.6)	7.6 (5.5–12.4)	2 (0.6–3.3)	5.3 (3.6–8.7)	1.4 (0.5–2.5)	26.5 (18.7–44.3)
No	455 (83.3)	44 (41–49)	2.4 (1.6–3.4)	4.2 (2.8–6.2)	7.4 (4.8–11.5)	1.8 (0.7–3)	5.1 (3.1–8.1)	1.4 (0.6–2.5)	25.2 (15.9–40.1)
**Pre-Pregnancy BMI**									
Underweight	14 (2.6)	46.5 (42–48.8)	2.5 (0.5–3.3)	5.6 (2.4–8.3)	11.2 (4.4–19.3)	2.7 (1.1–4.1)	7.9 (3.6–10.9)	2.4 (1.2–3.9)	39.5 (15–61.2)
Normal	332 (60.8)	44 (41–49)	2.5 (1.6–3.6)	4.6 (3.1–6.4)	8.1 (5.6–12.3)	2.1 (1.1–3.3)	5.7 (4–9.1)	1.7 (0.7–2.7)	27.3 (18.2–42.5)
Overweight	112 (20.5)	44 (41–48)	2.5 (1.6–3.4)	4.3 (2.9–6.4)	7.4 (4.5–11.8)	1.6 (0.7–2.9)	4.7 (3–7.8)	1.2 (0.5–2.5)	24.9(14.7–41.5)
Obese	88 (16.1)	44 (41–51)	2.2 (1.7–2.9)	3 (2.4–4.7)	5.2 (3.8–8)	0.8 (0.3–1.8)	3 (2.2–4.8)	1 (0.3–1.6)	17 (13.2–27.2)

^1^ Sum of PCBs 118, 138, 153, 170, 180, and 187 weighted by molar mass.

**Table 4 ijerph-16-00457-t004:** Bayesian Predictive Odds Ratios (BPORs) for the relation between mother PCB levels (quartiles) and child autistic behaviours defined by an SRS > 60 threshold, in MIREC study participants, Canada, 2008–2011 (*n* = 546).

			Adjusted Odds Ratio (95% CI)			
			Bayesian Results		Traditional Frequentist Results	
PCB Category ^1^	Value (ng/g Lipid)	*n*	BPOR ^2^	Probability OR > 1	Logistic Regression ^3^	OR for ASD in Lyall et al. [21] ^4^
**PCB118**						
Q1	<1.4	108	1.0 (referent)	0%	1.0	1.0
Q2	1.4-<2.3	143	0.93 (0.57, 1.44)	38%	1.57 (0.27, 11.3)	1.29 (0.86, 1.95)
Q3	2.3-<3.6	170	1.00 (0.62, 1.53)	50%	0.49 (0.07, 3.74)	1.38 (0.90, 2.11)
Q4	≥3.6	125	1.20 (0.72, 1.89)	77%	NA ^5^	1.15 (0.72, 1.82)
**PCB138**						
Q1	<3.2	175	1.0	0%	1.0	1.0
Q2	3.2-<5.5	184	1.21 (0.79, 1.76)	82%	3.10 (0.53, 28.0)	1.39 (0.92, 2.10)
Q3	5.5-<8.9	118	1.36 (0.84, 2.09)	91%	NA ^5^	1.34 (0.87, 2.07)
Q4	≥8.9	69	1.76 (0.99, 2.92)	98%	NA ^5^	1.79 (1.10, 2.92)
**PCB153**						
Q1	<4.2	87	1.0	0%	1.0	1.0
Q2	4.2-<7.4	178	1.36 (0.80, 2.16)	89%	1.98 (0.27, 41.4)	1.32 (0.88, 1.99)
Q3	7.4-<11.7	144	1.09 (0.62, 1.78)	63%	0.19 (0.01, 5.90)	1.24 (0.80, 1.93)
Q4	≥11.7	137	1.82 (1.02, 3.02)	98%	0.19 (0.01, 6.50)	1.82 (1.10, 3.02)
**PCB170**						
Q1	<1.5	227	1.0	0%	1.0	1.0
Q2	1.5-<2.6	141	0.90 (0.60, 1.31)	30%	0.46 (0.08, 2.11)	1.15 (0.76, 1.76)
Q3	2.6-<4.3	110	1.04 (0.65, 1.58)	57%	NA^5^	1.17 (0.75, 1.83)
Q4	≥4.3	68	1.39 (0.80, 2.24)	90%	0.30 (0.01, 2.71)	1.48 (0.88, 2.50)
**PCB180**						
Q1	<3.4	154	1.0	0%	1.0	1.0
Q2	3.4-<6.1	182	0.63 (0.40, 0.96)	19%	0.33 (0.06, 1.78)	1.00 (0.66, 1.50)
Q3	6.1-<10.4	120	0.79 (0.46, 1.24)	18%	0.11 (0.00, 1.10)	1.17 (0.75, 1.81)
Q4	≥10.4	90	1.20 (0.67, 1.98)	75%	0.14 (0.01, 1.58)	1.49 (0.89, 2.49)
**PCB187**						
Q1	<0.92	197	1.0	0%	1.0	1.0
Q2	0.92-<1.8	124	0.92 (0.60, 1.34)	62%	0.60 (0.10, 2.95)	0.89 (0.58, 1.36)
Q3	1.8-<3.3	135	0.99 (0.65, 1.44)	48%	0.23 (0.02, 1.42)	1.22 (0.79, 1.87)
Q4	≥3.3	90	1.46 (0.89, 2.24)	95%	NA ^5^	1.32 (0.79, 2.20)
**Sum of above PCBs**						
Q1	<33.4	358	1.0	0%	1.0	1.0
Q2	33.4-<55.3	110	1.32 (0.88, 1.92)	92%	0.32 (0.02, 2.16)	1.08 (0.72, 1.63)
Q3	55.3-<86.3	51	1.44 (0.82, 2.36)	91%	NA ^5^	0.99 (0.64, 1.51)
Q4	≥86.3	27	1.97 (0.90, 3.77)	97%	NA ^5^	1.36 (0.88, 2.11)

^1^ The Q1, Q2, Q3, Q4 are the 1st, 2nd, 3rd or 4th PCB quartiles from Table 1 of Lyall et al. [21]. ^2^ BPORs for autistic behaviour in MIREC using an SRS threshold of 60. Adjusted for child’s sex, mother’s age, race, marital status, education level, annual income, whether the mother has ever smoked during pregnancy, has ever consumed alcohol during pregnancy, and pre-pregnancy bmi. ^3^ Frequentist logistic regression using the dichotomized SRS data as the dependent variable (SRS > 60). Adjusted for child’s sex, mother’s age, race, marital status, education level, annual income, whether the mother has ever smoked during pregnancy, has ever consumed alcohol during pregnancy, and pre-pregnancy BMI. ^4^ ORs for ASD copied directly from Table 1 of Lyall et al. [21]. ^5^ Maximum likelihood estimator of the odds ratio did not converge.

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
