# Peer review of "Assessing the Relation between Plasma PCB Concentrations and Elevated Autistic Behaviours using Bayesian Predictive Odds Ratios"

_ijerph, 2019, doi:10.3390/ijerph16030457_

Round 1

Reviewer 1 Report

The authors conduct an analysis of prenatal PCB levels and autistic behaviors as assessed by the SRS-2. The topic is of high relevance and current interest, the authors use rigorous and relatively novel (for the field of autism) statistical methods, and I believe the paper is a worthwhile contribution. However, I have specific suggestions and comments that would clarify the work:

1.    Introduction: The recent follow up (Brown et al) to the FIPS pilot study (Cheslack-Postava et al) could be added to the brief review of the literature- no associations with PCBs were seen in the larger sample from the same study, demonstrating further lack of consistency and clarity in existing findings on the topic. 

2.    Methods: For PCB170 and 187, there is a high proportion below the LOD. The cut-off typically used has been at least 60% above the LOD – it may be helpful to justify this cut-off a bit. Also, the primary method for handling individuals with values <LOD for these congeners is not detailed. The language in the sensitivity analyses makes it sound as though imputed values were only used in those analyses – so were only individuals with detectable levels used in primary analyses? This needs to be more clearly laid out.  

3.    Methods: It may help motivate this choice of methodology (given it is not the “traditional” approach) by spelling out the benefits of this Bayesian approach more fully and earlier on in the paper– e.g., ability to examine all PCB congeners in a single model, thereby accounting for potential confounding by other PCBs when examining the effect estimate for a single congener. I think this approach could also be “sold” in the introduction as well. (Also, you note the method doesn’t account for mixtures, but there are Bayesian models to do this. Why not use an alternate approach, then, to address that?)

4.    Methods: Also regarding sensitivity analysis – I’m not clear on the model adjusting for plasma lipid concentration – do you mean instead of using the ng/g lipid values, using the raw with lipid as a covariate only, or adjusting the ng/g concentrations with total lipid in the model as well? I realize a reference is cited for further details on this approach, but think it would be helpful to spell out the motivation for and meaning of this a bit more.

5.    Methods: Given you are comparing results using a cut-off value of the SRS to those based on clinically-confirmed ASD diagnosis (e.g., in the EMA study from Lyall et al) – it is may be useful to consider more fully how well the SRS cut-off is corresponding to ASD diagnosis. I was going to suggest adding an analysis using a cut-off of 75 instead, until I reached the comment in the discussion that the BPOR is invariant to choice of cut-point. This is an important point that should be noted in your methods when the cut-off of >60 is introduced. Also, I think it needs to be clearly (can be brief) spelled out/ demonstrated why these BPOR are not impacted by the choice of cut-off. 

6.    Methods: Can you conduct cubic spline analyses in this Bayesian framework? My concern is with modeling only in quartiles, more complex non-linear associations may be missed.

7.    Results: Because one of the motivating threads throughout is comparison to existing/prior studies, in addition to noting how levels compare to Canadian levels overall, it would be helpful to note how they compare to US (could use NHANES) and European or Finnish levels (briefly work in to 170-176). Alternatively, move all comparisons to others populations to the Discussion and include this broader range.

8.    Results: If you are going to show the ORs from Lyall et al in the primary results table, the corresponding ng/g lipid concentrations for each quartile from this study need to also be shown – otherwise it is hard to judge the comparability of estimates since quartiles are not uniform across study populations. The catch is if plasma and serum concentrations are not directly comparable, some conversion may be needed and this should be noted in the text as well.

9.    Discussion: I find the whole concept of the ‘little things matter’ finding impactful, interesting, and important. I think some aspect of this translation of small finding into big impact should be added to that first paragraph.

10.  Despite the above point, it may also worth noting that the key strengths of the SRS are in its ability to quantitativelycapture broader ASD-related traits across the population. I would also modify the wording in this section – particularly on line 302 – suggest changing “it also measures other aspects” to “it also may capture other aspects” – given the high comorbidity between ASD and ADHD, and overlap in latent traits of these conditions, the fact that individuals with commonly comorbid conditions tend to score higher than unaffected individuals (though, on average, still below ASD case means) is not necessarily unexpected or a limitation. Also- the authors may comment on how this approach would compare to a study with both a quantitative outcome as well as (via a different measure) ASD diagnosis – and how/whether your study have been strengthened by having the ability to compare findings to those based off of ASD diagnosis according to clinical assessment, ADI-R, or ADOS-? 

11.  I would consider rewording how you present the main conclusions both in the first paragraph as well as the last – for example, stating in the first sentence you found no associations and then in the second you found some evidence is a little contradictory, and could be smoothed out, perhaps in a single sentence. 

12. Finally, in comparing to other work, it may be useful to note that in addition to differences in statistical methodology, differences in mixtures to which study populations are exposed could also account for discrepancies across existing work. (ie, if there are other environmental factors - be it the mix of other chemicals acting as confounders, or dietary factors that modify associations that vary across populations- this could be a source of variation impacting results as well).

Author Response

See attached pdf.

Reviewer 2 Report

Thank you for giving me the opportunity to review the manuscript “ Assessing the Relation Between Plasma PCB  concentrations and Elevated Autistic Behaviours  using Bayesian Predictive Odds Ratios”.

The topic of the manuscript is of great interest and the paper is generally well written and organized.

Please find below several comments which in my opinion can improve the paper:

1)     In the sentence “A total of 546 met all the above criteria for inclusion in our analysis (Table 1).” I would add the total population from 546 mother-child pairs were selected.

2)     Tables 1 and 2 should be under the results not methods part of the manuscript

3)       Some more details can be added into the description of not obvious covariates – how they were described/measured and categorized (education, income, marital status, BMI etc.)

4)     I would add subheadings into the results part of the manuscript – it makes it easier to understand and follow.

5)     Please delete the tables numbers from the discussion part of the manuscript (like in line 256).

Author Response

See attached PDF
